# Dynamic physical examination indicators of cardiovascular health: A single-center study in Shanghai, China

Rongren Kuang[1], Yiling Liao[1], Xinhan Xie[2], Biao Li[1], Xiaojuan Lin[1], Qiang Liu[1]*, Xiang Liu[3]*, Wenya Yu[4]*

1 Department of Cardiovascular Disease, Shenzhen Traditional Chinese Medicine Hospital, The Fourth Clinical Medical College of Guangzhou University of Chinese Medicine, Shenzhen, China, 2 Guangzhou University of Chinese Medicine, Guangzhou, China, 3 Department of Respiratory Disease, The 903rd Hospital of the People's Liberation Army, Hangzhou, China, 4 School of Public Health, Shanghai Jiao Tong University School of Medicine, Shanghai, China

* jsjyyuwenya@sina.cn (WY); immunol_liuxiang@sina.com (XL); 13802263916@139.com (QL)

**Data Availability Statement:** All relevant data are within the paper and its Supporting Information files.

## Abstract

Dynamic physical examination data can provide both cross-sectional and time-series characteristics of cardiovascular health. However, most physical examination databases containing health and disease information have not been fully utilized in China. Hence, this study aimed to analyze dynamic physical examination indicators for cardiovascular health to provide evidence for precise prevention and control of cardiovascular diseases in the primary prevention domain among healthy population with different demographic characteristics in Shanghai. Three-year continuous data were collected from the physical examination center of a hospital in Shanghai from 2018 to 2020, which included a total of 14,044 participants with an average age of 46.51±15.57 years. The cardiovascular status of overall healthy individuals may have a decreasing trend, which is manifested as a significant year-on-year decrease in high-density lipoprotein cholesterol; a significant year-on-year increase in total cholesterol, low-density lipoprotein cholesterol, and blood glucose levels; and a possible increasing trend of diastolic blood pressure, body mass index, and triglycerides. Healthy population with different sex and age groups have various sensitives to cardiovascular physical examination indicators. To conduct more accurate cardiovascular health management and health promotion for key populations in primary prevention, focusing on the dynamic trends of blood pressure, blood lipids, blood glucose, and body mass index in men and changes in total cholesterol in women over time is especially important. The age group of 50–69 years is key for better prevention and control of cardiovascular health.

## Introduction

Cardiovascular disease (CVD) poses a significant threat to the health of Chinese people, and its impact on health is associated with socioeconomic development and lifestyle changes [1]. Owing to the risks associated with CVDs, prevention and control of CVDs among healthy

**Funding:** This work was supported by the Soft Science Project of Shanghai Science and Technology Innovation Action Plan, grant number 21692191300 (receiver: W.Y.), the Zhejiang Provincial Natural Science Foundation of China, grant number LQ21H100001 (receiver: X.L.), and the Hangzhou Health Science and Technology Program, grant number B20220431 (receiver: X. L.). The funders had no role in study design, data collection and analysis, decision to publish, or preparation of the manuscript.

**Competing interests:** The authors have declared that no competing interests exist.

individuals is extremely important. Increased awareness of the importance of regular physical examinations among Chinese people provides favorable conditions for the primary prevention of CVDs in healthy individuals based on the dynamic changes in physical examination indicators. Moreover, increasing number of employers in China provide free annual physical examinations for their employees. However, healthy individuals with varying demographic characteristics might have different trends in cardiovascular health over time, and most physical examination databases containing health and disease information through time have not been fully utilized in China [2,3]. Thus, it increases the difficulty for health providers to provide precise primary prevention for CVDs in healthy people through monitoring of these dynamic indicators.

Although many health providers and scholars have acknowledged the importance of data from physical examination and that studies using data from different regions and populations have been conducted [4–7], these studies have some limitations. Physical examination data are longitudinal monitoring data, which not only include cross-sectional data characteristics but also important time-series effects for evaluating the trend of CVDs among healthy individuals. However, most existing studies in China analyze data from a cross-sectional perspective, without considering the important effect of time [8–14]. Moreover, studies based on small single-center samples introduce biases [15–17], adding to the challenges for health providers in preventing and controlling CVDs.

Therefore, this study aimed to conduct a larger sample analysis based on the cross-sectional and time-series characteristics of dynamic physical examination indicators to provide more accurate evidence for health providers to prevent and control CVDs in healthy individuals in Shanghai, China, especially in high-risk populations.

## Methods

### Study design

Data were extracted from the physical examination database of a physical examination center of a top tertiary hospital in Shanghai from 2018 to 2020. To ensure the three-year continuous data of healthy people, all physical examinations enrolled in this study were provided by participants' employers, which are usually on an annual basis. Healthy individuals in this study refers to the participants that initially has no systemic diseases, and were particularly free of CVDs. The data in this study include basic information of the participants, such as sex, age, time of physical examination, and results of physical examination indicators. The inclusion criteria were as follows: complete physical examination records from the physical examination center from 2018 to 2020 and complete results of the following indicators: systolic blood pressure (SBP), diastolic blood pressure (DBP), body mass index (BMI), total cholesterol (TC), triglycerides (TG), high-density lipoprotein cholesterol (HDL-C), low-density lipoprotein cholesterol (LDL-C), and blood glucose (Glu). The exclusion criteria were physical examination records of only one or two years and incomplete indicator results; as well as participants who developed CVDs between 2018 and 2020.

### Statistical analysis

Statistical analyses were performed using PASW Statistics for Windows, version 18·0 (SPSS Inc., Chicago, IL, USA). Categorical data are described as frequencies (%), and the measurement data are presented as mean (standard deviation) or median (interquartile range). To further explore the influence of time on cardiovascular physical examination indicators, variance analysis of repeated measurements was used to examine the effects of time and demographic factors. All tests were two-sided, and $p < 0.05$ was considered significant.

### Ethics statement

This study was approved by the Ethics Committee of Shanghai Jiao Tong University School of Medicine School of Public Health. Patient consent was waived by the Ethics Committee of Shanghai Jiao Tong University School of Medicine School of Public Health due to anonymity and the research risk is less than the minimum risk. All the data in this study were anonymous, and no personal information was used for analysis.

## Results

### Demographic characteristics

A total of 14,044 eligible participants were included in this study, most of whom were women (59.0%). The average age was 46.51±15.57 years in 2018. Most participants (28.4%) were aged 30–39 years, followed by those aged 40–49 and 50–59 years (Table 1).

### Characteristics of cardiovascular health indicators

The eight cardiovascular health (CVH) indicators were SBP, DBP, BMI, and levels of TC, TG, HDL-C, LDL-C, and Glu (Table 2). The overall mean value of each indicator for each year was within the reference range.

### Variance analysis results of repeated measurements

**Sex-based analysis.** Differences in SBP, DBP, BMI, and levels of TC, TG, HDL-C, LDL-C, and Glu in participants of different sexes from 2018 to 2020 were all significant ($p < 0.05$). Specifically, men had higher SBP, DBP, BMI, and levels of TG, LDL-C, and Glu than women, while their TC and HDL-C levels were lower. Based on sex, differences in the other seven indicators, except for SBP between each year, were significant. The results of pairwise comparisons suggest that the differences in both DBP and BMI values between each year from 2018 to 2020 were significant, with the highest value in 2019 and the lowest value in 2018. The differences in TC, LDL-C, and Glu levels between each year from 2018 to 2020 were significant, and the levels of these three indicators continued to increase. The difference in HDL-C levels between each year from 2018 to 2020 was significant, although the levels continued to decrease. The difference in TG levels was only significant between 2018 and 2019, with higher levels in 2019 (Table 3).

**Table 1. Demographic characteristics of participants.**

| Characteristics | N (%) / $\bar{x} \pm SD$ |
| --- | --- |
| Sex | |
| Male | 5760 (41.0%) |
| Female | 8284 (59.0%) |
| Age (years)* | 46.51±15.57 |
| 18–29 | 1734 (12.30%) |
| 30–39 | 3982 (28.40%) |
| 40–49 | 3211 (22.90%) |
| 50–59 | 1859 (13.20%) |
| 60–69 | 1822 (13.00%) |
| ≥70 | 1436 (10.20%) |

*Age of participants in 2018.

**Table 2. Characteristics of cardiovascular health indicators.**

| Indicators* | Mean | Standard deviation | Median | Interquartile range |
|---|---|---|---|---|
| SBP (2018) | 126.77 | 19.28 | 125.00 | 113.00~137.00 |
| SBP (2019) | 126.68 | 19.37 | 125.00 | 113.00~138.00 |
| SBP (2020) | 126.64 | 19.35 | 125.00 | 113.00~137.00 |
| DBP (2018) | 74.71 | 11.45 | 74.00 | 67.00~82.00 |
| DBP (2019) | 75.37 | 11.34 | 75.00 | 67.00~82.00 |
| DBP (2020) | 75.09 | 11.08 | 75.00 | 67.00~82.00 |
| BMI (2018) | 22.97 | 3.24 | 22.90 | 20.70~24.80 |
| BMI (2019) | 23.07 | 3.25 | 23.00 | 20.80~24.90 |
| BMI (2020) | 23.04 | 3.23 | 23.00 | 20.80~24.90 |
| TC (2018) | 4.87 | 0.91 | 4.80 | 4.24~5.41 |
| TC (2019) | 4.93 | 0.92 | 4.86 | 4.30~5.47 |
| TC (2020) | 4.99 | 0.93 | 4.90 | 4.35~5.55 |
| TG (2018) | 1.36 | 1.14 | 1.09 | 0.74~1.64 |
| TG (2019) | 1.38 | 1.12 | 1.09 | 0.75~1.66 |
| TG (2020) | 1.38 | 1.18 | 1.10 | 0.73~1.65 |
| HDL-C (2018) | 1.38 | 0.33 | 1.35 | 1.13~1.57 |
| HDL-C (2019) | 1.37 | 0.32 | 1.37 | 1.15~1.54 |
| HDL-C (2020) | 1.36 | 0.33 | 1.36 | 1.13~1.55 |
| LDL-C (2018) | 2.94 | 0.79 | 2.91 | 2.40~3.42 |
| LDL-C (2019) | 3.01 | 0.76 | 3.01 | 2.53~3.42 |
| LDL-C (2020) | 3.03 | 0.78 | 3.03 | 2.52~3.49 |
| Glu (2018) | 5.31 | 1.02 | 5.10 | 4.80~5.50 |
| Glu (2019) | 5.41 | 1.13 | 5.20 | 4.90~5.50 |
| Glu (2020) | 5.51 | 1.14 | 5.30 | 4.93~5.70 |

*SBP, systolic blood pressure; DBP, diastolic blood pressure; BMI, body mass index; TC, total cholesterol; TG, triglycerides; HDL-C, high-density lipoprotein cholesterol; LDL-C, low-density lipoprotein cholesterol; Glu, blood glucose.

**Age-based analysis.** Differences in SBP, DBP, BMI, and levels of TC, TG, HDL-C, LDL-C, and Glu of participants in different age groups from 2018 to 2020 were all statistically significant ($p < 0.05$). Specifically, SBP and Glu levels increased with age. DBP increased with age <50 years, reached its highest at age 50–69 years, and then decreased. BMI values and TC and TG levels increased with age <60 years, reached their highest levels at age 60–69 years, and then decreased. HDL-C level was the highest at age 18–29 years, followed by 40–49 years, 30–39 years, >70 years, and 60–69 years, with the lowest level at age 50–59 years. LDL-C level increased with age <50 years, reached its highest level at age 50–59 years, and then decreased. Based on age attributes, differences in the other seven indicators, except for SBP between each year, were significant. The pairwise comparisons indicated that differences in both DBP and BMI between each year from 2018 to 2020 were significant, with the highest values in 2019 and the lowest values in 2018. The differences in TC and Glu levels between each year from 2018 to 2020 were significant, and the levels continued to increase. The difference in TG was only significant between 2018 and 2019, with higher levels in 2019. The differences in HDL-C and LDL-C levels were significant between 2018 and 2019 and between 2018 and 2020, with the highest HDL-C levels and the lowest LDL-C levels in 2018 (Table 4).

**Table 3. Characteristics of cardiovascular health indicators of participants with different sexes.**

| Indicators | $\bar{x} \pm SD$ | Sex ($\bar{x} \pm SD$) | | t value | p value |
| --- | --- | --- | --- | --- | --- |
| | | Male | Female | | |
| SBP | | | | 590.312 | <0.0001 |
| 2018 | 126.77±15.57 | 131.12±17.79 | 123.74±19.70 | | |
| 2019 | 126.68±19.28 | 131.12±17.71 | 123.60±19.87 | | |
| 2020 | 126.64±19.37 | 130.57±17.78 | 123.91±19.93 | | |
| F value | 1.191 | | | | |
| p value | 0.304 | | | | |
| DBP | | | | 1438.597 | <0.0001 |
| 2018 | 74.71±11.45[a] | 78.35±11.34 | 72.19±10.83 | | |
| 2019 | 75.37±11.34[b] | 79.15±11.13 | 72.74±10.73 | | |
| 2020 | 75.09±11.08[c] | 78.56±10.84 | 72.68±10.60 | | |
| F value | 35.896 | | | | |
| p value | <0.0001 | | | | |
| BMI | | | | 1939.599 | <0.0001 |
| 2018 | 22.97±3.24[a] | 24.28±3.05 | 22.06±3.05 | | |
| 2019 | 23.07±3.25[b] | 24.38±3.11 | 22.16±3.02 | | |
| 2020 | 23.04±3.23[c] | 24.36±3.07 | 22.12±3.02 | | |
| F value | 35.912 | | | | |
| p value | <0.0001 | | | | |
| TC | | | | 71.734 | <0.0001 |
| 2018 | 4.87±0.91[a] | 4.81±0.88 | 4.90±0.93 | | |
| 2019 | 4.93±0.92[b] | 4.86±0.86 | 4.98±0.95 | | |
| 2020 | 4.99±0.93[c] | 4.90±0.88 | 5.05±0.96 | | |
| F value | 211.218 | | | | |
| p value | <0.0001 | | | | |
| TG | | | | 976.488 | <0.0001 |
| 2018 | 1.36±1.14[a] | 1.68±1.46 | 1.13±0.78 | | |
| 2019 | 1.38±1.12 | 1.69±1.36 | 1.17±0.86 | | |
| 2020 | 1.38±1.18 | 1.68±1.37 | 1.17±0.97 | | |
| F value | 4.079 | | | | |
| p value | 0.017 | | | | |
| HDL-C | | | | 3719.540 | <0.0001 |
| 2018 | 1.38±0.33[a] | 1.20±0.26 | 1.50±0.32 | | |
| 2019 | 1.37±0.32[b] | 1.21±0.26 | 1.48±0.30 | | |
| 2020 | 1.36±0.33[c] | 1.19±0.26 | 1.48±0.31 | | |
| F value | 20.123 | | | | |
| p value | <0.0001 | | | | |
| LDL-C | | | | 27.015 | <0.0001 |
| 2018 | 2.94±0.79[a] | 2.99±0.77 | 2.91±0.80 | | |
| 2019 | 3.01±0.76[b] | 3.04±0.74 | 2.99±0.78 | | |
| 2020 | 3.03±0.78[c] | 3.06±0.76 | 3.01±0.79 | | |
| F value | 162.779 | | | | |
| p value | <0.0001 | | | | |
| Glu | | | | 277.66 | <0.0001 |
| 2018 | 5.31±1.02[a] | 5.47±1.20 | 5.20±0.86 | | |
| 2019 | 5.41±1.13[b] | 5.58±1.32 | 5.29±0.96 | | |
| 2020 | 5.51±1.14[c] | 5.69±1.32 | 5.38±0.97 | | |

(*Continued*)

**Table 3.** (Continued)

| Indicators | $\bar{x} \pm SD$ | Sex ($\bar{x} \pm SD$) | | t value | p value |
|:---:|:---:|:---:|:---:|:---:|:---:|
| | | Male | Female | | |
| F value | 481.775 | | | | |
| p value | <0.0001 | | | | |

[a]$p < 0.05$, 2018 vs. 2019

[b]$p < 0.05$, 2019 vs. 2020

[c]$p < 0.05$, 2018 vs. 2020.

SBP, systolic blood pressure; DBP, diastolic blood pressure; BMI, body mass index; TC, total cholesterol; TG, triglycerides; HDL-C, high-density lipoprotein cholesterol; LDL-C, low-density lipoprotein cholesterol; Glu, blood glucose.

## Discussion

In contrast to previous studies, this study focused on healthy individuals with different demographic characteristics and the influence of both health indicators and time. The results of dynamic physical examination indicators from 2018 to 2020 suggest that the CVH status of healthy individuals in Shanghai demonstrated a decreasing trend, which is manifested as a significant decrease in HDL-C level; a significant increase in TC, LDL-C, and Glu levels; and a possible increasing trend of DBP, BMI, and TG level. The key for prevention and control of CVDs based on dynamic physical examination indicators for heathy individuals with different sex varies, and the most important age group is 50–69 years.

Based on the effects of sex and time, the development of various CVH indicators in men and the variation trend of TC in women should be the focus. Previous studies have reported that increasing SBP, DBP, BMI, and TG, LDL-C, and Glu levels and decreasing HDL-C levels increases the risk of CVDs [11,15,18–27]. Combined with the results of this study, the SBP, DBP, BMI, and TG, LDL-C, and Glu levels of men were higher than those of women for each year [28], while their HDL-C levels were lower than those of women. Therefore, these findings will provide explicit evidence for early identification and timely intervention in high-risk male populations based on the changing trend of the indicators mentioned above.

Given the more serious issues regarding the lifestyle and behavioral risk factors of CVDs in men, more reasonable and acceptable interventions on eating habits, physical activities, and smoking habits among men should be considered [21,29,30]. However, special attention should be paid to body weight control because large weight fluctuations would, in turn, increase the risk of CVDs [31,32]. Although the CVH indicators for women are better than those of men, evidence indicates that cardiovascular risk factors are often under-recognized and under-treated in women [33,34]. Furthermore, most existing studies have focused on female-specific factors (such as menopause and pregnancy-related issues) [35], while other factors remain neglected. Therefore, the TC levels of women should be considered by physicians. The growing trend in TC levels of women suggests that clinical intervention in marginal and high-risk populations, as well as increased regular follow-ups, are necessary to lower the risk of CVDs among women. Moreover, the quality of lifestyle-modifying interventions should be considered by healthcare providers because of the negative impact of large fluctuations [36].

Based on the effects of age and time, the impact of aging on CVH should be given importance [35,37–39], especially for those aged 50–69 years. First, advancing age increases CVD risks [40]. This study indicates that SBP and Glu level increase with age, which further increases the risk of CVDs among older individuals. Similar results were found by studies on the Japanese population, which demonstrated an increase in CVD risk with higher SBP in the

Table 4. Characteristics of cardiovascular health indicators of participants in different age groups.

| Indicators* | $\bar{x} \pm SD$ | Age ($\bar{x} \pm SD$), years | | | | | | t value | p value |
|---|---|---|---|---|---|---|---|---|---|
| | | 18–29 | 30–39 | 40–49 | 50–59 | 60–69 | ≥70 | | |
| SBP | | | | | | | | 1600.771 | <0.0001 |
| 2018 | 126.77±15.57 | 118.42±13.57 | 118.85±14.29 | 122.60±16.18 | 129.17±17.07 | 138.41±18.76 | 150.22±20.29 | | |
| 2019 | 126.68±19.28 | 117.50±13.49 | 118.57±14.72 | 122.47±15.78 | 129.50±16.87 | 138.97±17.87 | 150.47±20.67 | | |
| 2020 | 126.64±19.37 | 116.85±13.50 | 118.02±14.17 | 122.62±15.56 | 129.19±16.99 | 140.15±17.62 | 150.92±19.76 | | |
| F value | 0.084 | | | | | | | | |
| p value | 0.920 | | | | | | | | |
| DBP | | | | | | | | 303.858 | <0.0001 |
| 2018 | 74.71±11.45[a] | 70.94±9.70 | 71.83±10.54 | 74.6±11.64 | 78.62±11.86 | 78.56±11.36 | 77.57±11.23 | | |
| 2019 | 75.37±11.34[b] | 70.88±9.57 | 72.45±10.62 | 75.41±11.52 | 79.21±11.36 | 79.91±10.72 | 78.03±11.18 | | |
| 2020 | 75.09±11.08[c] | 70.49±9.26 | 72.23±10.49 | 75.38±11.09 | 78.69±11.02 | 79.49±10.75 | 77.67±10.85 | | |
| F value | 28.354 | | | | | | | | |
| p value | <0.0001 | | | | | | | | |
| BMI | | | | | | | | 78.127 | <0.0001 |
| 2018 | 22.97±3.24[a] | 21.78±3.48 | 22.70±3.40 | 23.04±3.10 | 23.48±2.91 | 23.59±2.92 | 23.55±3.10 | | |
| 2019 | 23.07±3.25[b] | 21.96±3.47 | 22.80±3.45 | 23.14±3.03 | 23.55±3.05 | 23.70±2.87 | 23.56±3.13 | | |
| 2020 | 23.04±3.23[c] | 22.06±3.57 | 22.82±3.42 | 23.10±3.05 | 23.44±2.89 | 23.64±2.89 | 23.41±3.16 | | |
| F value | 30.927 | | | | | | | | |
| p value | <0.0001 | | | | | | | | |
| TC | | | | | | | | 232.394 | <0.0001 |
| 2018 | 4.87±0.91[a] | 4.49±0.83 | 4.66±0.80 | 4.90±0.84 | 5.17±0.91 | 5.20±0.96 | 4.99±1.02 | | |
| 2019 | 4.93±0.92[b] | 4.56±0.80 | 4.72±0.81 | 4.96±0.83 | 5.23±0.96 | 5.27±0.98 | 5.06±1.06 | | |
| 2020 | 4.99±0.93[c] | 4.66±0.85 | 4.79±0.82 | 5.03±0.87 | 5.25±0.97 | 5.31±1.01 | 5.08±1.06 | | |
| F value | 200.435 | | | | | | | | |
| p value | <0.0001 | | | | | | | | |
| TG | | | | | | | | 145.889 | <0.0001 |
| 2018 | 1.36±1.14[a] | 0.96±0.70 | 1.19±1.06 | 1.38±1.32 | 1.62±1.14 | 1.68±1.33 | 1.53±0.86 | | |
| 2019 | 1.38±1.12 | 1.00±0.76 | 1.19±0.95 | 1.40±1.23 | 1.62±1.16 | 1.73±1.27 | 1.60±1.21 | | |
| 2020 | 1.38±1.18 | 1.03±0.91 | 1.22±1.24 | 1.37±1.10 | 1.61±1.29 | 1.69±1.35 | 1.53±0.88 | | |
| F value | 6.651 | | | | | | | | |
| p value | 0.001 | | | | | | | | |
| HDL-C | | | | | | | | 30.951 | <0.0001 |
| 2018 | 1.38±0.33[a] | 1.44±0.33 | 1.38±0.32 | 1.39±0.34 | 1.33±0.32 | 1.34±0.32 | 1.35±0.34 | | |
| 2019 | 1.37±0.32 | 1.43±0.31 | 1.37±0.31 | 1.38±0.32 | 1.32±0.32 | 1.33±0.30 | 1.35±0.32 | | |
| 2020 | 1.36±0.33[c] | 1.43±0.33 | 1.37±0.32 | 1.38±0.34 | 1.32±0.32 | 1.33±0.30 | 1.34±0.33 | | |
| F value | 22.149 | | | | | | | | |
| p value | <0.0001 | | | | | | | | |
| LDL-C | | | | | | | | 139.036 | <0.0001 |
| 2018 | 2.94±0.79[a] | 2.66±0.78 | 2.82±0.71 | 2.97±0.75 | 3.18±0.80 | 3.17±0.83 | 2.98±0.88 | | |
| 2019 | 3.01±0.76 | 2.76±0.69 | 2.89±0.69 | 3.03±0.70 | 3.23±0.82 | 3.23±0.82 | 3.04±0.83 | | |
| 2020 | 3.03±0.78[c] | 2.81±0.74 | 2.93±0.71 | 3.08±0.75 | 3.23±0.82 | 3.20±0.81 | 3.01±0.84 | | |
| F value | 135.911 | | | | | | | | |
| p value | <0.0001 | | | | | | | | |
| Glu | | | | | | | | 436.159 | <0.0001 |
| 2018 | 5.31±1.02[a] | 4.90±0.49 | 5.03±0.63 | 5.25±0.93 | 5.52±1.22 | 5.72±1.24 | 5.95±1.36 | | |
| 2019 | 5.41±1.13[b] | 4.96±0.40 | 5.08±0.65 | 5.36±1.10 | 5.62±1.33 | 5.87±1.36 | 6.12±1.57 | | |
| 2020 | 5.51±1.14[c] | 5.01±0.46 | 5.16±0.70 | 5.45±1.05 | 5.72±1.21 | 6.00±1.38 | 6.30±1.62 | | |

(Continued)

**Table 4.** (Continued)

| Indicators* | $\bar{x} \pm SD$ | Age ($\bar{x} \pm SD$), years | | | | | | t value | p value |
|---|---|---|---|---|---|---|---|---|---|
| | | 18–29 | 30–39 | 40–49 | 50–59 | 60–69 | ≥70 | | |
| F value | 500.647 | | | | | | | | |
| p value | <0.0001 | | | | | | | | |

[a] $p < 0.05$ 2018 *v.s.* 2019

[b] $p < 0.05$ 2019 *v.s.* 2020

[c] $p < 0.05$ 2018 *v.s.* 2020.

*SBP, systolic blood pressure; DBP, diastolic blood pressure; BMI, body mass index; TC, total cholesterol; TG, triglycerides; HDL-C, high-density lipoprotein cholesterol; LDL-C, low-density lipoprotein cholesterol; Glu, blood glucose.

older adult population [41,42]. Studies in China and India suggested the increasing trend of Glu level with age [43,44]. Therefore, to control the CVD risk induced by these two abnormal indicators, the management of blood pressure and Glu should be intensified by interventions such as exercise, diet, and lifestyle [45]. Second, the age group of 50–69 years is key for better prevention and control of CVH, and a general deteriorating trend in the CVH indicators of people in this age group can be observed. Although higher levels of DBP increases CVD risks, studies from various countries (e.g., the United States, Japan, and Asian-Pacific regions) suggest that the effect of DBP on CVH is relatively weaker than SBP, that is the effect of DBP among older-aged population is not as significant as relatively younger people [41,42,46,47]. Our study has a similar finding, suggesting that DBP increases with age <50 years and reaches its highest level at age 50–69 years. In addition, similar age-related trends were observed in indicators of LDL-C, BMI, TC, and TG. LDL-C increases with age <50 years and reaches its highest level at the age of 50–59 years, which was supported by a large-scale study on age-related LDL-C trends in the general Chinese population, of whom, the LDL-C starts to decrease from the age of 57 years[48]. BMI and TC and TG levels increase with age <60 years and reach the highest levels at the age of 60–69 years. Similar inverted U-shaped quadratic trajectories of these indicators with aging were observed in previous studies although the exact cut-off varied. A study in Korea demonstrated the inverted U-shaped trend between BMI and age [49]. A study in Netherland showed that TC decreases with age over 75 years [50], and a study in the United States indicated the cut-off was 50 years [51]. Various studies showed that the TG level increased until middle age (e.g., early 50s) and then showed subsequent decline [52–55]. Thereby, physicians should encourage people aged 50–69 years to increase the monitoring of CVH indicators and the frequency of follow-ups.

This study has some limitations. First, this study was single-centered owing to data accessibility, which may have led to some constraints in the representation and feasibility of our findings. Second, the data in this study only included indicators from physical examination items; thus, other indicators that are significant for the control and prevention of CVDs were not included. Third, the data in this study were only laboratory testing indicators, and other factors such as dietary habits, exercise habits, and sleep characteristics were not analyzed.

## Conclusions

The CVH status of healthy individuals in Shanghai demonstrated a decreasing trend from 2018 to 2020. Strengthening the primary prevention of CVD-related physical examination indicators among high-risk individuals, especially for the dynamic change in trends of SBP, DBP, BMI, and levels of TG, LDL-C, and Glu in men, as well as the change in trend of TC levels in women, is necessary. Moreover, healthcare providers should closely monitor the CVH

indicators in people aged 50–69 years, as these indicators are more sensitive to controlling the prevalence of CVDs in this population.

## Supporting information

**S1 Dataset.**
(XLSX)

## Author Contributions

**Conceptualization:** Rongren Kuang, Qiang Liu, Xiang Liu, Wenya Yu.

**Data curation:** Wenya Yu.

**Formal analysis:** Rongren Kuang, Xiang Liu.

**Funding acquisition:** Wenya Yu.

**Writing – original draft:** Rongren Kuang.

**Writing – review & editing:** Yiling Liao, Xinhan Xie, Biao Li, Xiaojuan Lin, Qiang Liu, Xiang Liu, Wenya Yu.

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
