## [Decision Letter · Decision Letter 0]

31 Mar 2022

PONE-D-22-07130Dynamic physical examination indicators of cardiovascular health: A single-center study in Shanghai, ChinaPLOS ONE

Dear Dr. Yu,

Thank you for submitting your manuscript to PLOS ONE. After careful consideration, we feel that it has merit but does not fully meet PLOS ONE’s publication criteria as it currently stands. Therefore, we invite you to submit a revised version of the manuscript that addresses the points raised during the review process.

We look forward to receiving your revised manuscript.

Kind regards,

Yajing Wang

Academic Editor

PLOS ONE

Journal Requirements:

Reviewers' comments:

Reviewer's Responses to Questions

**Comments to the Author**

1. Is the manuscript technically sound, and do the data support the conclusions?

Reviewer #1: Partly

Reviewer #2: Yes

2. Has the statistical analysis been performed appropriately and rigorously? 

Reviewer #1: Yes

Reviewer #2: Yes

3. Have the authors made all data underlying the findings in their manuscript fully available?

Reviewer #1: Yes

Reviewer #2: Yes

4. Is the manuscript presented in an intelligible fashion and written in standard English?

Reviewer #1: Yes

Reviewer #2: Yes

5. Review Comments to the Author

Reviewer #1: The authors of the present manuscript highlight the importance of the full utilization of physical examination databases in China. They performed a prospective study including 14,044 participants and a three-year follow-up. Overall, this manuscript is well prepared and organized. My major concerns are as listed as below:

1. The definition of healthy people is unclear. People with routine physical examinations are not qualified enough to be healthy. Accordingly, for the inclusion criterion, the authors should clearly mention that all the participants are initially without systemic diseases, especially are free of cardiovascular disease (CVD). If they didn’t strictly follow this criterion, they better reanalyze data (excluding individuals with diseases) or use a more accurate word to describe participants instead of “healthy individuals”.

2. They didn’t mention whether these participants developed new-onset CVD or with CVD-targeted medication treatments during the three-year period, which has a strong effect on these indicators they collected. Ignoring these factors may result in incorrect conclusions. If all the participants were finally free of CVD, in the exclusion criterion section, they should mention people developing CVD are excluded; Otherwise, they better analyze the percentage of people developing CVD and with CVD-targeted medication treatments. If these data are unavailable to the authors, they better explain them in the discussion part.

3. The age-based analysis of blood pressure and lipids levels are kinds of inconsistent with the major publications, and the references the authors referred to for this part are not convinced enough. The authors better do a deeper discussion about the potential factors or increase the inclusion criterion to get a more accurate conclusion.

Reviewer #2: 1. Incremental findings: Previous studies have reported that increasing SBP, DBP, BMI, and TG, LDL-C, and Glu levels and decreasing HDL-C levels increases the risk of CVDs. Combined with the results of this study, the SBP, DBP, BMI, and TG, LDL-C, and Glu levels of men were higher than those of women for each year, while their HDL-C levels were lower than those of women. Therefore, these findings will provide explicit evidence for early identification and timely intervention in high-risk male populations based on the changing trend of the indicators mentioned above.

2. Dietary habits, exercise habits, and sleep characteristics were not enrolled.

3. I suggest do analysis between male and female group.

4. There is Meta-analyzed publish related to this paper, suggest add to discussion.

5. The manuscript, particularly the results, need to be better organized to improve the readability.

6. PLOS authors have the option to publish the peer review history of their article (what does this mean?). If published, this will include your full peer review and any attached files.

---

## [Author Response · Author response to Decision Letter 0]

8 Apr 2022

RESPONSE TO REVIEWERS

Reviewer #1

The authors of the present manuscript highlight the importance of the full utilization of physical examination databases in China. They performed a prospective study including 14,044 participants and a three-year follow-up. Overall, this manuscript is well prepared and organized. My major concerns are as listed as below:

Q1. The definition of healthy people is unclear. People with routine physical examinations are not qualified enough to be healthy. Accordingly, for the inclusion criterion, the authors should clearly mention that all the participants are initially without systemic diseases, especially are free of cardiovascular disease (CVD). If they didn’t strictly follow this criterion, they better reanalyze data (excluding individuals with diseases) or use a more accurate word to describe participants instead of “healthy individuals”.

Response: Thank you for your valuable comments. We apologize for the unclear statement. To properly describe healthy individuals in our study and the reasons for inclusion and exclusion, we have defined the term and added explanations in the methods section. 

Page 4; lines 72-76

To ensure the three-year continuity of the data of healthy people, details of physical examinations in this study were obtained from the annual physical examinations of the enrolled participants, which were provided by the participants’ employers. Healthy individuals in this study refers to the participants that initially has no systemic diseases, and were particularly free of CVDs.

Page 4; lines 82-84

The exclusion criteria were physical examination records of only one or two years and incomplete indicator results; as well as participants who developed CVDs between 2018 and 2020.

Q2. They didn’t mention whether these participants developed new-onset CVD or with CVD-targeted medication treatments during the three-year period, which has a strong effect on these indicators they collected. Ignoring these factors may result in incorrect conclusions. If all the participants were finally free of CVD, in the exclusion criterion section, they should mention people developing CVD are excluded; Otherwise, they better analyze the percentage of people developing CVD and with CVD-targeted medication treatments. If these data are unavailable to the authors, they better explain them in the discussion part.

Response: All the participants were finally free of CVDs, which has been added as an exclusion criterion in the Methods section.

Page 4; lines 82-84

The exclusion criteria were physical examination records of only one or two years and incomplete indicator results; as well as participants who developed CVDs between 2018 and 2020.

Q3. The age-based analysis of blood pressure and lipids levels are kinds of inconsistent with the major publications, and the references the authors referred to for this part are not convinced enough. The authors better do a deeper discussion about the potential factors or increase the inclusion criterion to get a more accurate conclusion.

Response: Thank you for your comment. The in-depth discussion about age-based analysis of blood pressure and lipids levels has been added, and more related references supporting our findings have been cited. Please see detailed revisions on pages 17-18; lines 189-216. 

#Reviewer 2

The study by Kuang et al. attempted to analyze dynamic physical examination indicators for cardiovascular health to provide evidence for the precise prevention and control of cardiovascular diseases in the primary prevention domain among healthy individuals with different demographic characteristics in Shanghai. and if so, to provide more accurate evidence for health providers to prevent and control CVDs in healthy individuals, especially in high-risk populations.

The authors reported that The CVH status of healthy individuals in Shanghai demonstrated a decreasing trend from 2018 to 2020. Strengthening the primary prevention of CVD-related physical examination indicators among high-risk individuals, especially for the dynamic change in trends of SBP, DBP, BMI, and levels of TG, LDL-C, and Glu in men, as well as the change in trend of TC levels in women, is necessary. Moreover, healthcare providers should closely monitor the CVH indicators in people aged 50–69 years, as these indicators are more sensitive to controlling the prevalence of CVDs in this population.

Q1. Incremental findings: Previous studies have reported that increasing SBP, DBP, BMI, and TG, LDL-C, and Glu levels and decreasing HDL-C levels increases the risk of CVDs. Combined with the results of this study, the SBP, DBP, BMI, and TG, LDL-C, and Glu levels of men were higher than those of women for each year, while their HDL-C levels were lower than those of women. Therefore, these findings will provide explicit evidence for early identification and timely intervention in high-risk male populations based on the changing trend of the indicators mentioned above. 

Response: Thank you for your kind comments.

Q2. Dietary habits, exercise habits, and sleep characteristics were not enrolled.

Response: Thank you for your comment. Due to the data availability, it is difficult to obtain these characteristics, and thus these were presented as a limitation of this study. 

Page 18; lines 217-222

This study has some limitations. First, this study was single-centered owing to data accessibility, which may have led to some constraints in the representation and feasibility of our findings. Second, the data in this study only included indicators from physical examination items; thus, other indicators that are significant for the control and prevention of CVDs were not included. Third, the data in this study were only laboratory testing indicators, and other factors such as dietary habits, exercise habits, and sleep characteristics were not analyzed.

Q3. I suggest do analysis between male and female group.

Response: Thank you for your suggestion. The variance analysis results of repeated measurements based on different sexes from 2018 to 2020 were conducted. Please see the detailed results on pages 7-8; lines 117-128, and Table 3.

Q4. There is Meta-analyzed publish related to this paper, suggest add to discussion. 

Response: Thank you for your suggestion. We have cited additional references supporting and interpreting our findings combined your valuable and other Reviewer’s comments. Please see detailed revisions in the Discussion section.

Q5. The manuscript, particularly the results, need to be better organized to improve the readability. 

Response: Thank you for your comment. We revised the part of results to improve the readability. Please see the revisions in the Results section.

---

## [Decision Letter · Decision Letter 1]

28 Apr 2022

Dynamic physical examination indicators of cardiovascular health: A single-center study in Shanghai, China

PONE-D-22-07130R1

Dear Dr. Yu,

We’re pleased to inform you that your manuscript has been judged scientifically suitable for publication and will be formally accepted for publication once it meets all outstanding technical requirements.

Kind regards,

Yajing Wang

Academic Editor

PLOS ONE

Additional Editor Comments (optional):

Reviewers' comments:

Reviewer's Responses to Questions

**Comments to the Author**

1. If the authors have adequately addressed your comments raised in a previous round of review and you feel that this manuscript is now acceptable for publication, you may indicate that here to bypass the “Comments to the Author” section, enter your conflict of interest statement in the “Confidential to Editor” section, and submit your "Accept" recommendation.

Reviewer #1: All comments have been addressed

Reviewer #2: All comments have been addressed

2. Is the manuscript technically sound, and do the data support the conclusions?

Reviewer #1: Yes

Reviewer #2: Yes

3. Has the statistical analysis been performed appropriately and rigorously? 

Reviewer #1: Yes

Reviewer #2: Yes

4. Have the authors made all data underlying the findings in their manuscript fully available?

Reviewer #1: Yes

Reviewer #2: Yes

5. Is the manuscript presented in an intelligible fashion and written in standard English?

Reviewer #1: Yes

Reviewer #2: Yes

6. Review Comments to the Author

Reviewer #1: The manuscript looks fine now, just with one minor error:

In Page 4; lines 74-76,

“Healthy individuals in this study refers to the participants that initially has no systemic diseases, and were particularly free of CVDs.” is supposed to be “Healthy individuals in this study refers to the participants that initially have no systemic diseases, and were particularly free of CVDs.”

Reviewer #2: Incremental findings: Previous studies have reported that increasing SBP, DBP, BMI, and TG, LDL-C, and Glu levels and decreasing HDL-C levels increases the risk of CVDs. Combined with the results of this study, the SBP, DBP, BMI, and TG, LDL-C, and Glu levels of men were higher than those of women for each year, while their HDL-C levels were lower than those of women. Therefore, these findings will provide explicit evidence for early identification and timely intervention in high-risk male populations based on the changing trend of the indicators mentioned above. The manuscript readability was improved.

7. PLOS authors have the option to publish the peer review history of their article (what does this mean?). If published, this will include your full peer review and any attached files.

Reviewer #1: **Yes: **Fujie Zhao

Reviewer #2: No

---

## [Editor Report · Acceptance letter]

4 May 2022

PONE-D-22-07130R1 

Dynamic physical examination indicators of cardiovascular health: A single-center study in Shanghai, China 

Dear Dr. Yu:

I'm pleased to inform you that your manuscript has been deemed suitable for publication in PLOS ONE. Congratulations! Your manuscript is now with our production department. 

Kind regards, 

on behalf of

Dr. Yajing Wang 

Academic Editor

PLOS ONE